# VARIANCE REGULARIZING ADVERSARIAL LEARNING

## ABSTRACT

We study how, in generative adversarial networks, variance in the discriminator's output affects the generator's ability to learn the data distribution. In particular, we contrast the results from various well-known techniques for training GANs when the discriminator is near-optimal and updated multiple times per update to the generator. As an alternative, we propose an additional method to train GANs by explicitly modeling the discriminator's output as a bi-modal Gaussian distribution over the real/fake indicator variables. In order to do this, we train the Gaussian classifier to match the target bi-modal distribution implicitly through meta-adversarial training. We observe that our new method, when trained together with a strong discriminator, provides meaningful, non-vanishing gradients.

## 1 INTRODUCTION

Generative adversarial networks (Goodfellow et al., 2014) are a framework for training a generator of some target (i.e., "real") distribution without explicitly defining a parametric generating distribution or a tractable likelihood function. Training the generator relies on a learning signal from a discriminator or discriminator, which is optimized on a relatively simple objective to distinguish between (i.e., classify) generated (i.e., "fake") and real samples. In order to match the true distribution, the generator parameters are optimized to maximize the loss as defined by the discriminator, which by analogy makes the generator and discriminator adversaries.

In recent years, GANs have attained strong recognition as being able to generate high-quality images with sharp edges in comparison to maximum-likelihood estimation-based methods (Dempster et al., 1977; Kingma & Welling, 2013; Salakhutdinov & Hinton, 2009; Berthelot et al., 2017; Zhao et al., 2016). Despite their recent successes, GANs can also be notoriously hard to train, suffering from collapse (i.e., mapping its noise to a very small set of singular outputs), missing modes of the real distribution (Che et al., 2017), and vanishing and/or unstable gradients (Arjovsky & Bottou, 2017). In practice, successful learning is highly reliant on hyperparameter-tuning and model choice, and finding architectures that work with adversarial learning objectives with any / all of the above problems can be challenging (Radford et al., 2015).

Many methods have been proposed to address learning difficulties associated with learning instability.

- The use of autoencoders or posterior models in the generator or discriminator. These have been shown help to alleviate mode collapse or stabilize learning (Larsen et al., 2015; Che et al., 2017; Dumoulin et al., 2017; Donahue et al., 2016).
- Regularizing the discriminator, such as with input noise (Arjovsky & Bottou, 2017), instance noise (Salimans et al., 2016), and gradient norm regularization (Roth et al., 2017).
- Alternate difference measures, such as integral probability metrics (IPMs, Sriperumbudur et al., 2009) metrics. The most well-known of these use a dual formulation of the Wasserstein distance. These are implemented via either weight clipping (Arjovsky et al., 2017) or gradient penalty (Gulrajani et al., 2017). These can yeild stable gradients, high-quality samples and work on a variety of architectures.

On this last point, it is unclear whether the metric or the associated regularization used to impose Lipschitz is important, as regularization techniques have also been shown to be effective with stabilizing learning in $f$-divergences (Roth et al., 2017). In this paper, we study the integral role of

variance in the discriminator's output in the regime of the generated distribution and how it ultimately affects learning. In the following sections, we describe theoretical motivations, an empirical analysis from multiple variants of GANs, and finally propose a regularization scheme to combat vanishing gradients on part of the discriminator when it is well-trained.

## 2 METHODS

The generative adversarial framework entails training a generating function $G_\theta : \mathcal{Z} \to \mathcal{X}$, where $\mathbf{z} \in \mathcal{Z}$ is noise sampled from a relatively simple prior (such as spherical Gaussian), $p(\mathbf{z})$, and $G_\theta$ is modeled by a deep neural network with parameters, $\theta$. The objective is to find the optimal parameters such that the induced or generated distribution, $\mathbb{Q}_\theta$, matches a desired target distribution, $\mathbb{P}$. In order to accomplish this, adversarial models define a *discriminator* function, $D_{\phi,d} : \mathcal{X} \to \mathbb{R}$, modeled by a deep neural network with parameters, $\phi$. The objective of the discriminator is to estimate a measure of distance or *divergence*, $d(\mathbb{P}, \mathbb{Q}_\theta)$, between the two distributions, $\mathbb{P}$ and $\mathbb{Q}_\theta$. In practice, which divergence is best to estimate in this context is still a subject of debate, but it suffices to say that the most common distances come from either the family of $f$-divergences (Nguyen et al., 2010; Nowozin et al., 2016), such as the Jensen-Shannon divergence (JSD) or $\chi$-squared divergence (Mao et al., 2016) or Integral Probability Metrics (IPMs, Sriperumbudur et al., 2009; Mroueh & Sercu, 2017), such as the Wasserstein distance (Arjovsky et al., 2017; Gulrajani et al., 2017) and maximum mean discrepancy (MMD, Sutherland et al., 2016; Li et al., 2017). Generally, the estimator given by the family of functions defined by $D_{\phi,d}$ represent a family of lower-bounds of the respective divergence:

$$d(\mathbb{P}, \mathbb{Q}_\theta) \geq V_d(D_{\phi,d}, G_\theta, \mathbb{P}). \tag{1}$$

In Goodfellow et al. (2014), $V_d$ is called the *value function*. Optimization of the discriminator function by maximizing the value function roughly corresponds to finding the supremum over all $D_{\phi,d}$. The GAN objective is the central quantity of interest in the adversarial saddle-point problem, given $\mathbb{P}$, $p(\mathbf{z})$, $G_\theta$, and $D_{\phi,d}$:

$$\min_\theta \max_\phi V_{JSD\star}(D_{\phi,d}, G_\theta, \mathbb{P}) = \min_\theta \max_\phi \mathbb{E}_{\mathbf{x}\sim\mathbb{P}}[\log D(\mathbf{x})] + \mathbb{E}_{\mathbf{x}\sim\mathbb{Q}_\theta}[\log(1 - D(\mathbf{x}))]. \tag{2}$$

where $V_{JSD\star}(D_{\phi,d}, G_\theta, \mathbb{P})$ is a lower bound to $JSD^\star = 2*JSD - 2\log 2$. Here, $D = \sigma \circ \phi$ [1] is the composition of the sigmoid function, $\sigma(y) = \frac{1}{1+exp(-y)}$, and a neural network, $\phi$. More generally, Nowozin et al. (2016) more formally define a GAN objective in terms of a family of $f$-divergences:

$$V_f(D_{\phi,d,\mathbb{P}}, G_\theta) = \mathbb{E}_{\mathbf{x}\sim\mathbb{P}}[\psi_f(\mathbf{x})] - \mathbb{E}_{\mathbf{x}\sim\mathbb{Q}_\theta}[f^\dagger(\psi_f(\mathbf{x}))], \tag{3}$$

where $f^\dagger$ is a convex conjugate to the $f$-divergence generator function and $\psi_f = g_f \circ \phi$ is the composition of a nonlinearity and neural network, such that $g_f$ respects the domain of the domain of the corresponding $f$-divergence.

### 2.1 STRONG OR WEAK METRIC?

The above family of $f$-divergence estimators such as the JSD one above are powerful and can produce generators with high quality samples on complex datasets (Radford et al., 2015), but there is some evidence that they provide poor learning signals for the generator parameters, $\theta$. As shown in Arjovsky & Bottou (2017), the JS divergence will be flat everywhere important if $\mathbb{P}$ and $\mathbb{Q}_\theta$ both lie on low-dimensional manifolds (as is likely the case with real data) and do not prefectly align. In principle, one would think that a better divergence estimate (tighter bound) to a strong divergence metric would provide the most meaningful learning signal for the generator. In practice, however, as we optimize the JSD discriminator, it will have zero gradients on $\mathcal{X}$ and hence provide almost no learning signal for the generator. In addition, the proxy loss, $\mathbb{E}_{\mathbf{x}\sim\mathbb{Q}_\theta}[\log D(\mathbf{x})]$, which is commonly used as an alternative to the GAN objective in training the discriminator can be highly unstable, and is generally plagued by missing modes and model collapse. Finally, as we will show below, other $f$-divergences (e.g., Mao et al., 2016) suffer different types of instabilities that can make training difficult to impossible.

---

[1]Here, we introduce a slight abuse of notation, using $\phi$ to represent a deep neural network and its parameters.

Some of the above observations motivate the use of a "weaker" metric that use weaker topologies than that of $f$-divergences, notably the earth-movers distance or *Wasserstein distance* (Arjovsky et al., 2017). Besides being a useful information metric for computing the distance between two distributions, the Wasserstein distance has the nice property of being differentiable nearly everywhere, and thus provides a better convergence properties on the generator. The Kantorovich-Rubinstein dual problem of estimating the Wasserstein distance involves the GAN objective:

$$W_1(\mathbb{P}, \mathbb{Q}_\theta) \geq V_W(D_{\phi,d,\mathbb{P}}, G_\theta) = \mathbb{E}_{\mathbf{x} \sim \mathbb{P}}\left[D(\mathbf{x})\right] - \mathbb{E}_{\mathbf{x} \sim \mathbb{Q}_\theta}\left[D(\mathbf{x})\right] : ||D||_L \leq 1, \tag{4}$$

where $D$ is constrained to be in the set of Lipschitz-1 functions [2]. This GAN objective is a lower-bound for the Wasserstein, as we saw with $f$-GANs and the $f$-divergences, and adversarial models that follow this metric are called Wasserstein GANs (WGAN Arjovsky et al., 2017). The Lipchitz constraint should ensure a smoothness or continuity or the gradients being given to the generator. In principle then, under this constraint, one can train the discriminator to optimality without having to worry about any catastrophic effects on the generator.

The difficulty, however, is in constraining $D$ to be Lipschitz, and it is conventional to use some sort of regularization (Arjovsky et al., 2017; Gulrajani et al., 2017). However, questions remain regarding said regularization and whether the supremum of the resulting family of functions is close enough to the true Wasserstein distance. In addition, there is strong evidence that regularization on $f$-GANs can improve training stability (Roth et al., 2017), indicating that having a looser bound over a constrained family of discriminator functions may in principle be a better strategy for training a GAN.

## 2.2 The Importance of Variance

Taking the perspective provided in the previous section, our goal becomes to train a discriminator to near-optimality in the hopes that this will improve learning in the generator. A well-known scenario is one in which the discriminator can perfectly discern samples from $\mathbb{P}$ from samples from $\mathbb{Q}_\theta$. Moreover, (Arjovsky & Bottou, 2017) show that the discriminator is likely to be constant in the regime of $\mathbb{Q}_\theta$, thus emitting zero gradients and no learning signal for the generator. Consider such a discriminator; if $\mathbf{x}, \mathbf{y} \sim \mathbb{Q}_\theta$ are samples, then as the generator attempts to minimize any discrepancy between $\mathbb{P}$ and $\mathbb{Q}_\theta$, the slope of the tangent hyperplane connecting $(\mathbf{x}, \phi(\mathbf{x}))$ and $(\mathbf{y}, \phi(\mathbf{y}))$ likely obeys

$$\frac{|\phi(\mathbf{x}) - \phi(\mathbf{y})|}{d_\mathcal{X}(\mathbf{x}, \mathbf{y})} = 0 \tag{5}$$

where $d_\mathcal{X}(\cdot, \cdot)$ is a metric defined on $\mathcal{X}$. This roughly corresponds to a flat discriminator over $\mathbb{Q}_\theta$ which doesn't emit any gradients and impedes the generator from learning. As a result, we find

$$\mathbb{E}_{\mathbf{x} \sim \mathbb{Q}_\theta}\left[\frac{\partial \phi}{\partial \mathbf{x}}(\mathbf{x})\right] = 0. \tag{6}$$

Alternatively, a flat discriminator in the regime of $\mathbb{Q}_\theta$ can be interpreted as *having minimal variance* (see Figure 1). That is, when the discriminator is mostly constant over $\mathbb{P}$ and $\mathbb{Q}_\theta$, the discriminator's unnormalized output distribution is peaked and tends to mimic a bi-modal Dirac distribution. This is because, in a sense, the discriminator acts as a bi-modal mapping: real examples from $\mathbb{P}$ are mapped to, for example, 1 while generated examples from $\mathbb{Q}_\theta(\mathbf{x})$ are mapped to 0. In contrast, when the discriminator is non-constant over the data/generated manifolds and exhibits variance in its output over within $\mathbb{P}$ and/or $\mathbb{Q}_\theta$, its unnormalized output distribution is spread out and not peaked.

## 2.3 Variance Regularizing GANs

Following these observations, we propose a regularization technique targeting the discriminator via *meta-adversarial learning*. We hypothesize that constraining the discriminator's output distribution

---

[2]Though the $K$-Lipschitz condition suffices, in practice, it is common to use 1-Lipschitz as this does not change the optimization in any meaningful way.

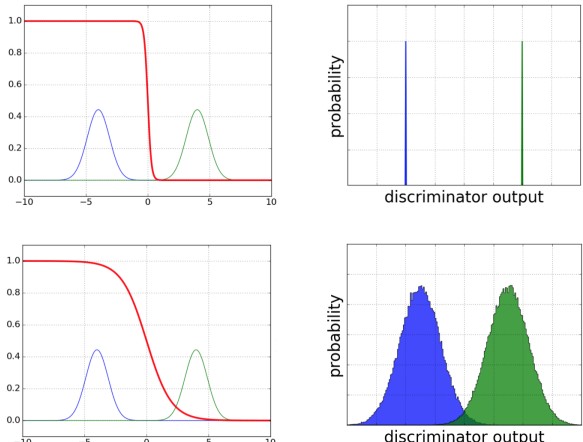

Figure 1: The red line represents the discriminator's output, the blue and green lines represent the real and fake distributions, respectively. **Top**: When the discriminator is near-perfect, it's unnormalized output distribution is peaked. **Bottom**: Variance in discriminator output within the data and/or generated distribution will result in variance in its unnormalized output distribution. Note that the discriminator's output need not be in the range $(0, 1)$, we provide just an example here.

to follow a mixture of Gaussians will have the same effect as the use of different metrics and/or regularization as seen in works discussed above.

Let $p(y|l = 0)$ and $p(y|l = 1)$ be two univariate Gaussian distributions with unit variance and means $\mu_f$ and $\mu_r$, respectively, and where the indicator variable as values, $l = 0$ that corresponds to "fake" or generated and $l = 1$ that corresponds to real. Define the mixture:

$$p_\phi^\star(y) = \frac{1}{2}(p(y|l = 0) + p(y|l = 1)). \tag{7}$$

Depending on the location of the means, this Gaussian classifier has an associated classification error that is directly related to their overlap. In principle, the classifier defines a bound on a difference or distance between two distributions defined by the values of the indicator variable, $l = 0$ and $l = 1$. As the discriminator is really a classifier that in principle is trying to minimize the Bayes risk, under the above constraint, the bound is maximized when the means, $\mu_f$ and $\mu_r$, are as far apart as possible. This is obviously undesirable as this can lead to the same problems mentioned in the previous sections. Here we will take a different strategy: keep the means and variance *fixed* so that the associated classifier has a fixed error rate. This will enforce a sort of fixed "weakness" on the discriminator, which corresponds to a fixed "looseness" on the lower bound defined by the classifier.

We ensure the discriminator's output follows $p_\phi^\star(y)$ through adding two additional discriminators which are trained in a similar was as the original GAN formulation:

$$\min_\phi \max_\mathcal{R} V_\mathcal{R}(\mathcal{R}, \phi) = \mathbb{E}_{\{y^{(i)}\}_{i=1}^m \sim N_r(y)} \left[ \log \mathcal{R}(\{y^{(i)}\}_i) \right] + \mathbb{E}_{\{\mathbf{x}^{(i)}\}_i \sim \mathbb{P}} \left[ \log \left( 1 - \mathcal{R}(\{\phi(\mathbf{x}^{(i)})\}_i) \right) \right] \tag{8}$$

$$\min_\phi \max_\mathcal{F} V_\mathcal{F}(\mathcal{F}, \phi) = \mathbb{E}_{\{y^{(i)}\}_{i=1}^m \sim N_f(y)} \left[ \log \mathcal{F}(\{y^{(i)}\}_i) \right] + \mathbb{E}_{\{\mathbf{x}^{(i)}\}_i \sim \mathbb{Q}_\theta} \left[ \log \left( 1 - \mathcal{F}(\{\phi(\mathbf{x}_i)\}_i) \right) \right] \tag{9}$$

where $\mathcal{F}$ and $\mathcal{R}$ are the are the fake and real *output discriminators*, respectively, each parametrized by a two-layer MLP. $\mathcal{F}$'s goal is to discern a batch of real values $\{y_i\}_i$ drawn from a Gaussian with mean $\mu_f$ and unit variance from a batch taken from the discriminator's output over fake samples. Likewise, $\mathcal{R}$ follows a similar protocol for real samples. It is important to note that comparisons are performed batch-wise rather than per sample. In essence, $\mathcal{F}$ and $\mathcal{R}$ are *meta-discriminators* which instill a regularizing effect on the discriminator.

Meanwhile, the original GAN formulation is only slightly perturbed, as the generator tries to match $\mathbb{Q}_\theta$ to $\mathbb{P}$ while the discriminator attempts to differentiate real and fake samples. The discriminator's

loss objective combines equations 8 and 9:

$$\mathcal{L}_\phi = -\mathbb{E}_{\{\mathbf{x}_i\}_{i=1}^m \sim \mathbb{Q}_\theta}[\log \mathcal{F}(\{\phi(\mathbf{x}^{(i)})\}_i)] - \mathbb{E}_{\{\mathbf{x}_i\}_{i=1}^m \sim \mathbb{P}}[\log \mathcal{R}(\{\phi(\mathbf{x}^{(i)})\}_i)]. \tag{10}$$

The generator's loss function is:

$$\mathcal{L}_{G_\theta} = \mathbb{E}_{\mathbf{x} \sim \mathbb{Q}_\theta}[\phi(\mathbf{x})] - \mathbb{E}_{\mathbf{x} \sim \mathbb{P}}[\phi(\mathbf{x})], \tag{11}$$

standard of IPMs. In practice, however, we find that driving the discriminator's output towards $\mu_r$ using a least squares loss also works well:

$$\mathcal{L}'_{G_\theta} = \mathbb{E}_{\mathbf{x} \sim \mathbb{Q}_\theta}\left[(\phi(\mathbf{x}) - \mu_r)^2\right]. \tag{12}$$

---

**Algorithm 1** Variance Regularizing Adversarial Learning (VRAL) using meta-discriminators. Default parameter values are $\alpha = 10^{-4}$, $\beta_1 = 0.5$, $\beta_2 = 0.999$, $\epsilon = 3 \times 10^{-6}$, $m = 64$. $n_{\text{batches}}$ is determined by the chosen dataset.

**Input:** $n_{\mathcal{F},\mathcal{R},\phi}$, the number of updates to perform on $\mathcal{F}$, $\mathcal{R}$ and $\phi$ per update to $G_\theta$.

1: **while** $\theta_G$ not converged **do**
2:     **for** $i = 1 \ldots n_{\text{batches}}$ **do**
3:         **for** $k = 1 \ldots n_{\mathcal{F},\mathcal{R},\phi}$ **do**

4:             Sample$\{y^{(i)}\}_{i=1}^m \sim N_f(y)$, $\{\mathbf{y}^{(i)}\}_{i=1}^m \sim \mathbb{Q}_\theta$
5:             $g_F \leftarrow \nabla_{\theta_\mathcal{F}} \frac{1}{m} \sum_{i=1}^m \left[\log \mathcal{F}(\{y^{(i)}\}_i) + \log(1 - \mathcal{F}(\{\phi(\mathbf{y}^{(i)})\}_i))\right]$
6:             $\theta_\mathcal{F} \leftarrow \theta_\mathcal{F} + \alpha \cdot \text{Adam}(g_\mathcal{F}, \beta_1, \beta_2, \epsilon)$

7:             Sample$\{y^{(i)}\}_{i=1}^m \sim N_r(y)$, $\{\mathbf{x}^{(i)}\}_{i=1}^m \sim \mathbb{P}$
8:             $g_R \leftarrow \nabla_{\theta_\mathcal{R}} \frac{1}{m} \sum_{i=1}^m \left[\log \mathcal{R}(\{y^{(i)}\}_i) + \log(1 - \mathcal{R}(\{\phi(\mathbf{x}^{(i)})\}_i))\right]$
9:             $\theta_\mathcal{R} \leftarrow \theta_\mathcal{R} + \alpha \cdot \text{Adam}(g_\mathcal{R}, \beta_1, \beta_2, \epsilon)$

10:           Sample$\{\mathbf{x}^{(i)}\}_{i=1}^m \sim \mathbb{P}$, $\{\mathbf{y}^{(i)}\}_{i=1}^m \sim \mathbb{Q}_\theta$
11:           $g_\phi \leftarrow \nabla_{\theta_\phi} \frac{1}{m} \sum_{i=1}^m \left[\log \mathcal{F}(\{\phi(\mathbf{y}^{(i)})\}_i) + \log \mathcal{R}(\{\phi(\mathbf{x}^{(i)})\}_i)]\right]$
12:           $\theta_\phi \leftarrow \theta_\phi + \alpha \cdot \text{Adam}(g_\phi, \beta_1, \beta_2, \epsilon)$

13:           Sample$\{\mathbf{x}^{(i)}\}_{i=1}^m \sim \mathbb{P}$, $\{\mathbf{y}^{(i)}\}_{i=1}^m \sim \mathbb{Q}_\theta$
14:           $g_G \leftarrow \nabla_{\theta_G} \frac{1}{m} \sum_{i=1}^m \left[\phi(\mathbf{y}^{(i)}) - \phi(\mathbf{x}^{(i)})\right]$
15:           $\theta_G \leftarrow \theta_G - \alpha \cdot \text{Adam}(g_G, \beta_1, \beta_2, \epsilon)$

---

The training procedure for our proposed GAN with regularizing networks is similar to standard GAN training, however requiring each of the meta-discriminators to be updated prior to the discriminator update. In practice, we update $\mathcal{F}$, $\mathcal{R}$ and $D$ the same $n_{\mathcal{F},\mathcal{R},D}$ number of times per generator update. The full training procedure is presented in algorithm 1.

Our current formulation of Variance Regularizing GANs aims to fix $\mathbb{E}_{\mathbf{x} \sim \mathbb{P}}[\phi(\mathbf{x})]$ and $\mathbb{E}_{\mathbf{x} \sim \mathbb{Q}_\theta}[\phi(\mathbf{x})]$ at values $\mu_r$ and $\mu_f$, respectively. However, this can become problematic during optimization as the discriminator's goal is to maximize the difference between these two means. To accommodate, we subtract the mean of the discriminator's output over a batch of fake samples. This useful insight results in the modified objective in which the discriminator's output over fake examples is no longer fixed to a distribution centered at $\mu_f$, but can move around and preserves unit variance. Consequently, each occurrence of $\{\phi(\mathbf{x}^{(i)})\}_i$ where $\mathbf{x} \sim \mathbb{Q}_\theta$ in equation 10 is replaced with $\{\phi(\mathbf{x}^{(i)})\}_i - \mathbb{E}_{\mathbf{x} \sim \mathbb{Q}_\theta}[\phi(\mathbf{x})]$. As we will show in subsequent sections, this modification proves extremely helpful.

## 2.4 META-ADVERSARIAL LEARNING

In future sections, we show our method which uses meta-discriminators to regularize the discriminator's output distribution to be successful. However, this same technique can be applied to encourage

GANs exhibit other desirable properties. To this end, we demonstrate how we can enforce the discriminator to approximate a $K$-Lipschitz function. Consider a similar setup to Variance Regularizing GANs in which only one meta discriminator $\mathcal{M}$ is used, and its objective is

$$
\begin{aligned}
\mathcal{L}_{\mathcal{M}} = - & \mathbb{E}_{\{y^{(i)}\}_i \sim \mathcal{N}(y; \mu^*, \sigma^*)} \left[ \log \mathcal{M}(\{y^{(i)}\}_i) \right] \\
- & \mathbb{E}_{\{\mathbf{x}^{(i)}\}_i \sim \mathbb{P}, \{\mathbf{y}^{(i)}\}_i \sim \mathbb{Q}_\theta} \left[ \log \left( 1 - \mathcal{M} \left( \left\{ \frac{|\phi(\mathbf{x}^{(i)}) - \phi(\mathbf{y}^{(i)})|}{d_{\mathcal{X}}(\mathbf{x}, \mathbf{y})} \right\}_i \right) \right) \right],
\end{aligned}
\tag{13}
$$

where $\mu^* = K$ and $\sigma^*$ can vary, but is typically close to 0 (e.g. 0.1). The discriminator's objective is almost identical to equation 10, but its goal is to now maximize the second term in the above equation. In essence, the single meta-discriminator $\mathcal{M}$ is distinguishing samples drawn from a Gaussian distribution fixed at $\mu^*$ versus the slope of the hyperplane between $\mathbf{x}$ and $\mathbf{y}$, where $\mathbf{x}$ and $\mathbf{y}$ are real and generated examples, respectively.

## 3 RELATED WORK

Linear Discriminant Analysis (LDA) is a commonly used technique for dimensionality reduction and/or binary classification. Consider two sets of classes, $\mathbb{P}$ and $\mathbb{Q}_\theta$, and a set of observations $\{\phi(\mathbf{x})\}$, where each $\mathbf{x} \in \mathcal{X}$ (i.e., $\phi(\mathbf{x})$ may be interpreted as the result of dimensionality reduction on $\mathbf{x}$). The objective is to preserve the discrepancy between classes while maintaining intra-class variance. Let $p_\phi(\phi(\mathbf{x}); \mathbf{x} \sim \mathbb{P})$ and $p_\phi(\phi(\mathbf{x}); \mathbf{x} \sim \mathbb{Q}_\theta)$ be the probability densities of each class, assumed to follow a Gaussian distribution with means $\mu_r$ and $\mu_f$ and variances $\sigma_r^2$ and $\sigma_f^2$, respectively. For the sake of simplicity, assume $\sigma_r^2 = \sigma_f^2 = 1$. The LDA classification rule is then to predict $\phi(\mathbf{x})$ belonging to the class of samples from $\mathbb{Q}_\theta$ if

$$
(\phi(\mathbf{x}) - \mu_r)^2 - (\phi(\mathbf{x}) - \mu_f)^2 > T,
\tag{14}
$$

otherwise $\mathbb{P}$, and $T$ is a chosen threshold. Notice that when $\phi(\mathbf{x})$ is near $\mu_r$, the LHS of equation 14 becomes small (assuming $\phi(\mathbf{x})$ is far from $\mu_f$) and vice-versa when $\phi(\mathbf{x})$ is close to $\mu_f$.

McGan (Mroueh et al., 2017) employs mean and covariance feature matching between $\mathbb{P}$ and $\mathbb{Q}_\theta$. To achieve this, a function $\Phi : \mathcal{X} \to \mathbb{R}^m$, parametrized as a deep neural network, which maps $\mathbf{x} \in \mathcal{X}$ to an embedding space is learned. The objective is to minimize the mean and covariance between $\Phi(\mathbb{P})$ and $\Phi(\mathbb{Q}_\theta)$. More recently, Fisher GANs (Mroueh & Sercu, 2017) constrain the second order *raw* moments of the discriminator such that $\frac{1}{2}(\mathbb{E}_{\mathbf{x} \sim \mathbb{P}}[\phi^2(\mathbf{x})] + \mathbb{E}_{\mathbf{x} \sim \mathbb{Q}_\theta}[\phi^2(\mathbf{x})]) = 1$ using Lagrange multipliers. This approach slightly differs from Variance-Regularizing GANs, which enforce that variance, the second order *central* moments of the discriminator, are equal to 1. Nonetheless, Fisher GANs are similar to Variance Regularizing GANs as they aim to stabilize variance in the discriminator's output, although not explicitly defining a distribution which it should follow.

## 4 EMPIRICAL RESULTS

### 4.1 GAN ARCHITECTURE

We run experiments on the CIFAR-10 (Krizhevsky & Hinton, 2009) and CelebA (Liu et al., 2015) datasets for image generation. We use a DCGAN (Radford et al., 2015) model in all experiments and perform all parameter updates using the Adam optimizer (Kingma & Ba, 2015). For our proposed method, the meta-discriminators $\mathcal{F}$ and $\mathcal{R}$ are MLPs with 28 hidden units with a sigmoid output.

### 4.2 ANALYSIS

Having established theoretical motivations for a well-trained discriminator, we show how many GAN algorithms fail to preserve variance in the discriminator output when trained at large ratios, such as 50 discriminator updates per generator update. We train the standard GAN, standard GAN using proxy loss (i.e., $-\log D$ loss), Least Squares GAN (Mao et al., 2016), Wasserstein GAN and our proposed method (both with and without subtracting means) on CIFAR-10 with a varying number of discriminator updates per generator update.

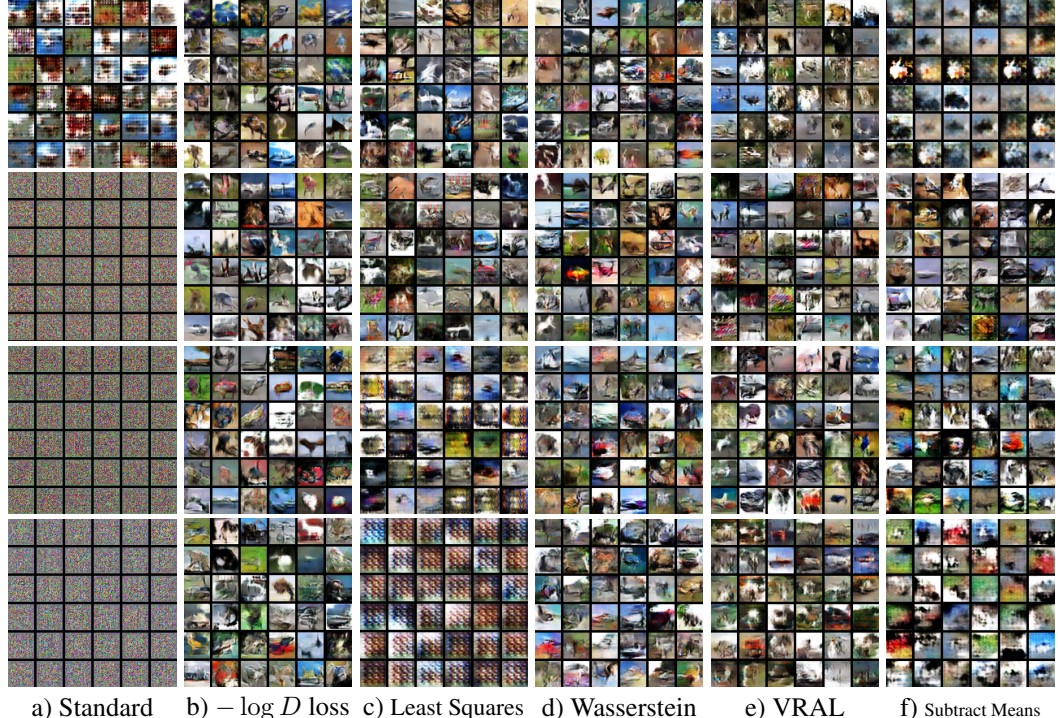

a) Standard    b) $-\log D$ loss  c) Least Squares  d) Wasserstein    e) VRAL    f) Subtract Means

Figure 2: Generated CIFAR-10 samples using the standard GAN, GAN with $-\log D$ loss, Least Squares GAN, Wasserstein GAN, and two versions of VRAL (with and without subtracting means) over various training ratios. The top row shows results from training each method at a 1:1 training ratio. Likewise, the second, third, and fourth shows show results from 10:1, 25:1 and 50:1 training. The standard and least squares GANs are not robust to such large training ratios while others continue to generate sharp images.

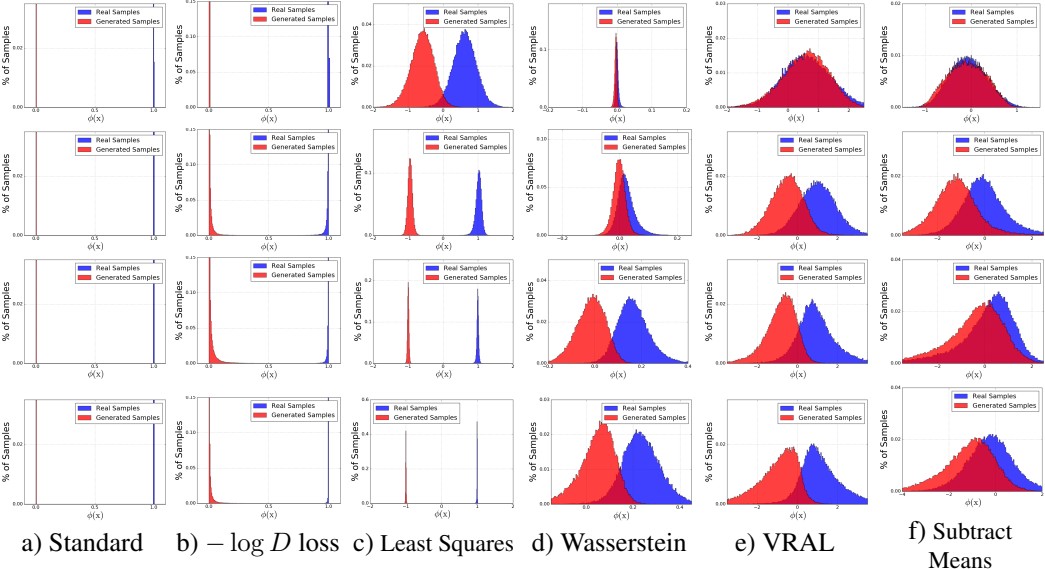

a) Standard    b) $-\log D$ loss  c) Least Squares  d) Wasserstein    e) VRAL    f) Subtract Means

Figure 3: Histograms depicting the discriminator's unnormalized output distribution for the standard GAN, GAN with $-\log D$ loss, Least Squares GAN, Wasserstein GAN and our proposed method when trained with various training ratios. In descending order, the results in each row correspond to 1:1, 10:1, 25:1 and 50:1 training ratios.

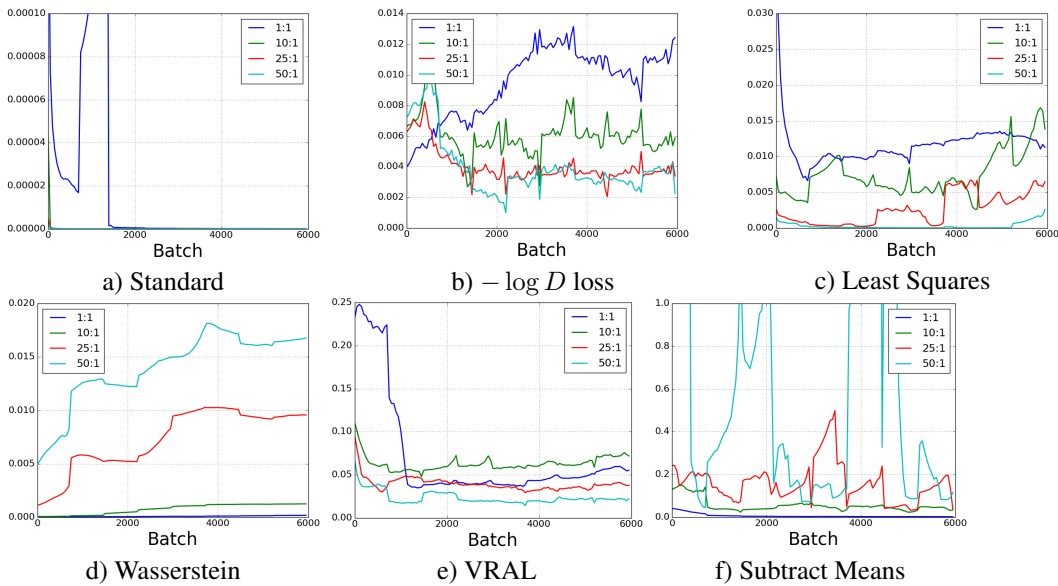

Figure 4: The discriminator's gradient norms w.r.t. generated samples during training for each method. Note that values are averaged over 50 batches.

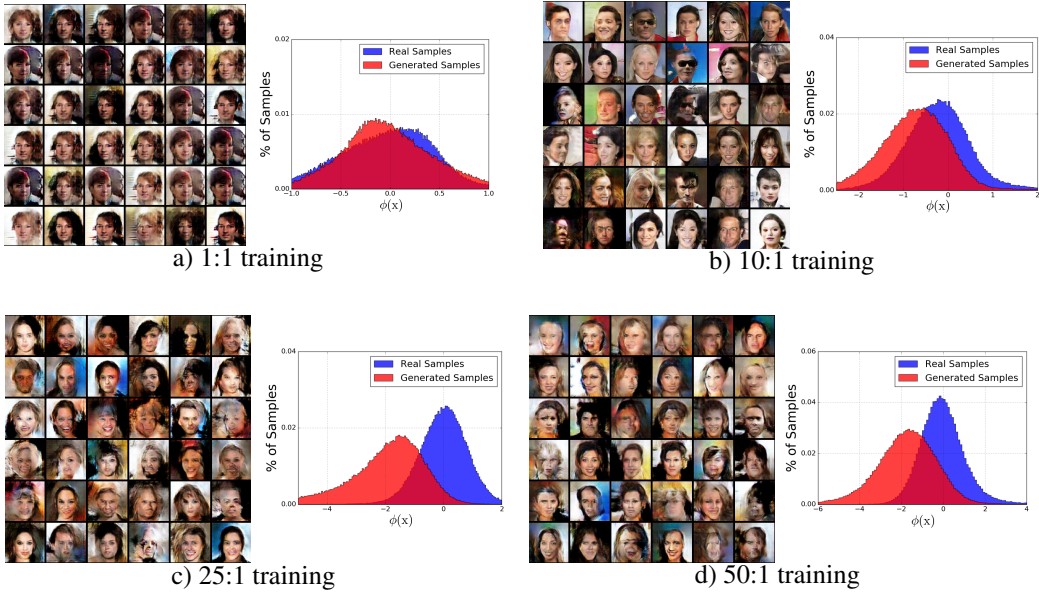

Figure 5: Samples and discriminator output histograms from training at various training ratios on CelebA.

We find that variance in the discriminator's unnormalized output distribution directly correlates with the generator's ability to learn if the discriminator is well-trained. Figure 2 shows samples from updating the discriminator 50 times per generator update (which we refer to as 50:1 training), while figure 3 demonstrates the discriminator's expected real-valued output over the training set. Interestingly, the discriminators in standard and Least Squares GANs exhibit almost no output variance within $\mathbb{P}$ or $\mathbb{Q}_\theta$, suggesting the discriminator is constant over these regimes in image space and would emit zero gradients w.r.t. any generated sample, hence hindering learning in the generator. Notice that these methods fail to learn at a 50:1 ratio Furthermore, the generated samples also posit that some amount of variance is necessary; GANs using $-\log D$ and Wasserstein losses are able

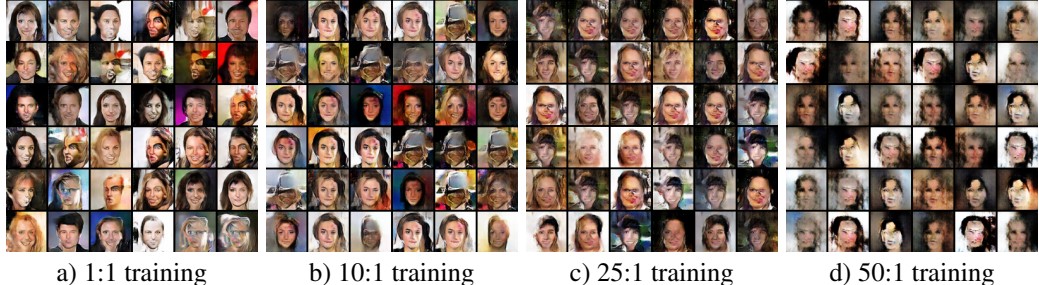

a) 1:1 training            b) 10:1 training            c) 25:1 training            d) 50:1 training

Figure 6: Generated images on CelebA by forcing the discriminator to be 1-Lipschitz using a single meta-discriminator.

to generate sharp samples while the discriminator in these instances is imperfect and exhibits some level of uncertainty amongst the generated samples.

Our hypothesis of (near) zero gradients is evidently supported as we contrast the discriminator's gradient norms for any particular GAN at varying update ratios. Figure 4 indicates that as the discriminator achieves perfection by being updated more frequently than the generator, $||\nabla_{\mathbf{x}\sim\mathbb{Q}_\theta}\mathcal{L}_\phi||$ approaches zero. However, the standard GAN with $-\log D$ loss and Wasserstein GAN are much more robust to large update ratios. The discriminators of these methods don't map all samples, real or fake, to a single value, hence providing a meaningful learning signal for the generator and avoiding the vanishing gradient scenario. Our proposed method follows this line of reasoning, as the meta-discriminators prevent the discriminator from achieving perfection, regardless of update ratio.

### 4.3 1-Lipschitz Discriminators

Adhering to our earlier discussion of using a single meta-discriminator to force the discriminator to approximate a $K$-Lipschitz function, we trained a meta-discriminator $\mathcal{M}$ to perform batch-wise discrimination between samples drawn from $\mathcal{N}(y; \mu^* = 1, \sigma^* = 0.1)$ and $\frac{|\phi(\mathbf{x})-\phi(\mathbf{y})|}{d_\mathcal{X}(\mathbf{x},\mathbf{y})}$ where $\mathbf{x} \sim \mathbb{P}$, $\mathbf{y} \sim \mathbb{Q}_\theta$. See figure 6 for results. More work is needed to robustly train GANs using meta-adversarial learning.

## 5 Conclusions

In this paper, we have demonstrated the importance of intra-class variance in the discriminator's output. In particular, our results show that methods whose discriminators tend to map inputs of a class to single real values are unable to provide a reliable learning signal for the generator. Furthermore, variance in the discriminator's output is essential to allow the generator to learn in the presence of a well-trained discriminator. We proposed a technique, conceptually in line with LDA, which ensures the discriminator's output distribution follows a specified prior. Taking a broader perspective, we also introduced a new regularization technique called meta-adversarial learning, which can be applied to ensure enforce various desirable properties in GANs.

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
