# OpenReview forum: "Variance Regularizing Adversarial Learning"
_ICLR.cc/2018/Conference — Reject_

### Official Review · AnonReviewer1 · 2017-11-25
**Interesting paper that needs more investigation**

**Rating:** 5
**Confidence:** 4

**Review:**

This paper studies how the variance of the discriminator affect the gradient signal provided to the generator and therefore how it might limit its ability to learn the true data distribution.

The approach suggested in this paper models the output of the discriminator using a mixture of two Gaussians (one for “fake” and the other for “not fake”). This seems like a rather crude approximation as the distribution of each “class” is likely to be multimodal. Can the authors comment on this? Could they extend their approach to use a mixture of multimodal distributions?

The paper mentions that fixing the means of the distribution can be “problematic during optimization as the discriminator’s goal is to maximize the difference between these two means.“. This relates to my previous comment where the distribution might not be unimodal. In this case, shifting the mean doesn’t seem to be a good solution and might just yield to oscillations between different modes. Can you please comment on this?

Mode collapse: Can you comment on the behavior of your approach w.r.t. to mode collapse?

Implementation details: How is the mean of the two Gaussians initialized?

Relation to instance noise and regularization techniques: Instance noise is a common trick being used to train GANs, see e.g. http://www.inference.vc/instance-noise-a-trick-for-stabilising-gan-training/
This also relates to some regularization techniques, e.g. Roth et al., 2017 that provides a regularizer that amounts to convolving the densities with white Gaussian noise. Can you please elaborate on the potential advantages of the proposed solution over these existing techniques?

Comparison to existing baselines: Given that the paper addresses the stability problem, I would expect some empirical comparison to at least one or two of the stability methods cited in the introduction, e.g. Gulrajani et al., 2017 or Roth et al., 2017.

Relation to Kernel MMD: Can the authors elaborate on how their method relates to approaches that replace the discriminator with MMD nets. e.g.
- Training generative neural networks via Maximum Mean Discrepancy optimization, Dziugaite et al
- Generative models and model criticism via optimized maximum mean discrepancy, Sutherland et al
More explicitly, the variance in these methods can be controlled via the bandwidth of the kernel and I therefore wonder what would one use a simple mixture of Gaussians instead?

---

### Official Review · AnonReviewer2 · 2017-11-28
**Interesting idea, but needs more work**

**Rating:** 4
**Confidence:** 4

**Review:**

The paper proposes variance regularizing adversarial learning (VRAL), a new method for training GANs.

The motivation is to ensure that the gradient for the generator does not vanish. The authors propose to use a discriminator whose output targets a mixture of two Gaussians (one component each for real and fake data).  The means and variances are fixed so that the discriminator does not overfit, which ensures that the generator learning is not hindered.

The discriminator itself is trained through two additional meta-discriminators (!) Are the meta-discriminators really necessary? Have you tried matching moments or using other methods for comparing the distributions?

It would be useful to write down the actual loss function so that it's easier to compare with other GAN variants. In particular, I'm curious to understand the difference between VRAL and Fisher-GAN. The authors discuss this in the end of Section 3, but a more careful comparison is needed.

The experimental results are pretty limited and lack detailed quantitative evaluation, which makes it harder to compare the performance of the proposed variant to existing algorithms.

Overall, I think that the idea is interesting, but the paper needs more work and does not meet the ICLR acceptance bar.

FYI, another concurrent submission showed that gradient penalties stabilize training of GANs:
MANY PATHS TO EQUILIBRIUM: GANS DO NOT NEED TO DECREASE A DIVERGENCE AT EVERY STEP
https://openreview.net/pdf?id=ByQpn1ZA-

---

### Official Review · AnonReviewer3 · 2017-11-28

**Rating:** 6
**Confidence:** 3

**Review:**

The authors provided empirical analysis of different variants of GANs and proposed a regularization scheme to combat the vanishing gradient when the discriminator is well trained.

More specifically, the authors demonstrated the importance of intra-class variance in the discriminator’s output. Methods whose discriminators tend to map inputs of a class to single real values are unable to provide a reliable learning signal for the generator, such as the standard GAN and Least Squares GAN. Variance in the discriminator’s output is essential to allow the generator to learn in the presence of a well-trained discriminator. To ensure the discriminator’s output follows the mixture of two univariate Gaussians, the authors proposed to add two additional discriminators which are trained in a similar was as the original GAN formulation. The technique is related to Linear Discriminant Analysis. From a broader perspective, the new meta-adversarial learning can be applied to ensure various desirable properties in GANs.

The performance of variance regularization scheme was evaluated on the CIFAR-10 and CelebA data.

Summary:
——
I think the paper discusses a very interesting topic and presents an interesting direction for training the GANs. A few points are missing which would provide significantly more value to readers. See comments below for details and other points.

Comments:
——
1.	Why would a bi-modal distribution be meaningful? Deep nets implicitly transform the data which is probably much more effective than using complex bi-modal Gaussian distribution; the bi-modal concept can likely be captured using classical techniques.

2.	On page 4, in Eq. (8) and (9), it remains unclear what $\mathcal{R}$ and $\mathcal{F}$ really are beyond two-layer MLPs; are the results of those two-layer MLPs used as the mean of a Gaussian distribution, i.e., $\mu_r$ and $\mu_f$?

3.	Regarding the description above Eq. (12), what is really used in practice, i.e., in the experiments? The paper omits many details that seem important for understanding. Could the authors provide more details on choosing the generator loss function and why Eq. (12) provides satisfying results in practice?

Minor Comments:
——
1.	In Sec 2.1, the sentence needs to be corrected: “As shown in Arjovsky & Bottou (2017), the JS divergence will be flat everywhere important if P and Q both lie on low-dimensional manifolds (as is likely the case with real data) and do not prefectly align.”

2.	Last sentence in Conclusion: “which can be applied to ensure enforce various desirable properties in GANs.” Please remove either “ensure” or “enforce.”

---

### Decision · Program_Chairs · 2018-01-29
**ICLR 2018 Conference Acceptance Decision**

**Decision:**

Reject

**Comment:**

The reviewers found a number of short-comings in this work that would prevent it from being accepted at ICLR in its current form, both in terms of writing (not specifying the loss function),  experiments that are too limited, and inconclusive comparisons with existing regularization techniques. I recommend the authors take into account the feedback from reviewers in any follow-up submissions.